# Impacts of Age, Genotype and Feeding Low-Protein Diets on the N-Balance Parameters of Fattening Pigs

**Ilona Anna Geicsnek-Koltay [1], Zsuzsanna Benedek [2], Nóra Hegedűsné Baranyai [3], Nikoletta Such [1], László Pál [1], László Wágner [1], Ádám Bartos [1], Ákos Kovács [2], Judit Poór [4] and Károly Dublecz [1,*]**

1. Institute of Physiology and Nutrition, Hungarian University of Agriculture and Life Sciences, Georgikon Campus, Deák Ferenc u. 16, 8360 Keszthely, Hungary; ilcsu92@gmail.com (I.A.G.-K.); such.nikoletta.amanda@uni-mate.hu (N.S.); pal.laszlo@uni-mate.hu (L.P.); wagner.laszlo@uni-mate.hu (L.W.); bartos.adam.sandor@uni-mate.hu (Á.B.)
2. Institute of Animal Sciences, Hungarian University of Agriculture and Life Sciences, Georgikon Campus, Deák Ferenc u. 16, 8360 Keszthely, Hungary; benedek.zsuzsanna@uni-mate.hu (Z.B.); kovacs.akos@uni-mate.hu (Á.K.)
3. Nagykanizsa University Center for Circular Economy, University of Pannonia, Zrínyi Miklós u. 18, 8800 Nagykanizsa, Hungary; baranyai.nora@uni-pen.hu
4. Institute of Mathematics and Basics of Natural Sciences, Hungarian University of Agriculture and Life Sciences, Georgikon Campus, Deák Ferenc u. 16, 8360 Keszthely, Hungary; poor.judit@uni-mate.hu
* Correspondence: dublecz.karoly@uni-mate.hu

**Abstract:** The effects of feeding low-protein (LP) diets and the age and genotype of fattening pigs were evaluated in an N-balance trial. Sixty weaned piglets of two genotypes were allotted to three different diets. Besides the control diets for the crossbred Topigs 20 × DanBred Duroc (TD) and Hungarian Large White (HLW) pigs, two LP diets were fed containing 1.5 (T1.5) and 3% (T3) less dietary protein than the control. The LP diets were supplemented with crystalline lysine, threonine, tryptophan, and methionine to equalize their digestible amino acid contents. Starter diets were fed between 20–30, grower I between 30–40, grower II between 40–80 and finisher between 80–110 kg live weights. Pigs were kept in floor pens, with 10 animals per pen. In all phases, six pigs with similar live weight were placed into individual balance cages and in the frame of a seven-day long balance trial, the daily N-intake, fecal and urinary N-excretion were measured. From the data N-digestibility, the total ammoniacal nitrogen (TAN) and N-retention were calculated. All the investigated main factors, the genotype and age of pigs and the protein content of the diets had significant effects on the N-balance of fattening pigs. The determinacy of the factors depended on the investigated parameter. Fecal N-excretion and N-digestibility were steadier compared with the urinary N-exertion and TAN percentage. N-digestibility increased and the urinary N-excretion decreased when LP diets were fed. The urinary N-decreasing effect of LP diets was not linear. Compared with the control (19.6 gN/day), T1.5 treatment resulted in 14.5, treatment T3 in 12.4 g daily urinary N-excretion. The TAN and the N-retention of HLW pigs were more favorable than those of TD pigs. Based on our results, it can be concluded that the accuracy of the nitrogen and TAN excretion values of pigs, used in the calculation of the national NH$_3$ inventories, could be improved if the genotype, the more detailed age categories and the different protein levels of feeds are considered.

**Keywords:** fattening pig; nitrogen balance; low protein diets; age; genotype

## 1. Introduction

The efficiency of protein utilization of farm animals is important because protein-rich feedstuffs are the most expensive components of diets, and the countries of the European Union are not self-sufficient.

Since the utilization of dietary protein affects the nitrogen excretion of farm animals, protein nutrition is closely related to the environmental aspects of animal production.

Ammonia emission is a major air quality concern at the regional, national, and global levels, and animal production is the main source of ammonia emission.

Because only 70–80% of the protein content of feed is absorbed from the digestive tract and the utilization of the absorbed fraction is even less than that of digestibility, a significant proportion of the feed protein consumed by farm animals is excreted in the form of urine and feces [1]. The metabolism and excretion of excess nitrogenous substances need extra energy, and from the undigested protein, bacteria in the hindgut produce significant amounts of potentially toxic compounds (phenols, amines, hydrogen sulfide, etc.) [2]. From an ammonia emission point of view, the urinary N is the main source of ammonia emission. It is the main part of the so-called total ammoniacal nitrogen (TAN) responsible for further ammonia formation [3,4]. From the urea and uric acid excreted via urine, urease-producing bacteria liberate ammonia in the manure. Ammonia has negative effects on the health of both animals and laborers in the barn, and it can impair even the production traits [5].

There are a wide range of published N-retention (31–45%), urinary (49–51%) and fecal (16–20%) nitrogen excretion values of fattening pigs [6]. Nitrogen excretion can be reduced by feeding low protein diets, applying more nutritional phases or using different feed additives, like exogenous enzymes [6]. Decreasing the crude protein content of diets can be achieved by using more crystalline amino acids. With this method, the crude protein content can be reduced by 2–3% without negative effects on the production traits [7]. According to the literature data, 1% protein reduction can decrease ammonia emissions by about 10% [7–9].

Additional feeding options that affect ammonia emissions are the modification of the urinary and fecal N ratio by feeding fermentable carbohydrates, using benzoic acid that reduces the pH of urine [6,9].

Today, in Europe the ammonia emission inventory is based on the different options of the EMEP/EEA Guidelines. The calculation uses the activity data of the countries (livestock, feed composition, etc.) and the default or country-specific N excretion and TAN emission factors. All the countries are interested in improving the accuracy of their calculations by producing country-specific data and emission factors [4].

There is quite a lot of information available on the effects of the above-mentioned nutritional techniques on the N-balance and ammonia emission of pigs. On the other hand, few data exist, such as how the age and genotype of fattening pigs affect the nitrogen excretion of animals if low-protein diets are fed.

## 2. Materials and Methods

The experiment was carried out in the pig research farm of the Institute of Physiology and Nutrition, Hungarian University of Agriculture and Life Sciences, Georgikon Campus, Keszthely, Hungary. The trial was approved by the Food Chain Safety and Land Office Department of the Zala County Government Office (case number: ZAI/040/01010-7/2018).

Altogether 60 weaned male piglets, 30–30 of two genotypes were selected to have similar live weight and placed into 6-floor pens of 10 pigs per pen. The size of pens was 3.5 × 3.4 m. Wheat straw was used as bedding material, and the pens were equipped with self-feeders and drinkers. The manure was removed daily.

Half of the piglets were Hungarian Large White (HLW), belonging to the late-maturing types of meat-type pigs, with high growth potential until a high body weight. The remaining 30 animals were crossbred pigs (Topigs 20 × Danbred Duroc) (TD) and represented the early maturing types of meat pigs, with more intensive growth potential and higher protein requirements in the early phases of fattening. Four phases fattening was carried out in the live weight categories of 20–30 kg, 30–40 kg, 40–80 kg and 80–110 kg. Starter, grower I, grower II and finisher diets were fed between days 53–71 and 72–80 and; 81–127 and 128–162, respectively. The diets of both groups were composed according to the requirements of the intensive crossbred (TD) and semi-intensive Hungarian Large White (HLW) genotypes [1]. Diets were formulated on a standardized ileal digestible (SID) amino acid basis. The SID amino acid contents of feedstuffs were evaluated by NIR. Besides a normal

crude protein-containing control diet (C), two low-protein (LP) diets were fed. The protein reductions in each phase were 1.5 (T1.5) and 3% (T3).

The composition and measured nutrient contents of experimental diets are shown in the Appendix A (Tables A1–A4). The diets contained maize, wheat and extracted soybean meal as main ingredients. All feeds contained crystalline lysine, methionine and threonine. Tryptophan supplementation was needed only in the low-protein diets. As the protein content of the diets reduced, the proportion of maize, sunflower oil and crystalline amino acids increased, while the amount of extracted soybean meal decreased. The proportion of soybean meal in T3 diets was on average 10% lower, compared with the control diets. The diets were formulated to be isocaloric, and the main difference in their nutrient content was only in crude protein.

In each age category, 6 pigs per pen with similar live weight were selected and transferred to a different room containing specific balance cages (Figure 1). The cages were equipped to collect separately the total amounts of feces. The amount of daily feed was calculated as 95% of the ad libitum feed intake. Daily ratios were distributed in two equal portions and given to the animals at 7.00 a.m. and 3.00 p.m. Water was provided ad libitum. In the balance cage room, heat blowers were used for heating and an exhaust air chimney for ventilation. The room temperature was $16 \pm 2\ ^{\circ}C$. Nine hours of the daily light period was applied with 80 lux. The $NH_3$ and $CO_2$ air concentrations were measured daily with Draeger equipment (Draeger x-am 5600). The ranges of $CO_2$ and $NH_3$ air concentrations were 400–1100 and 0–2.6 ppm, respectively.

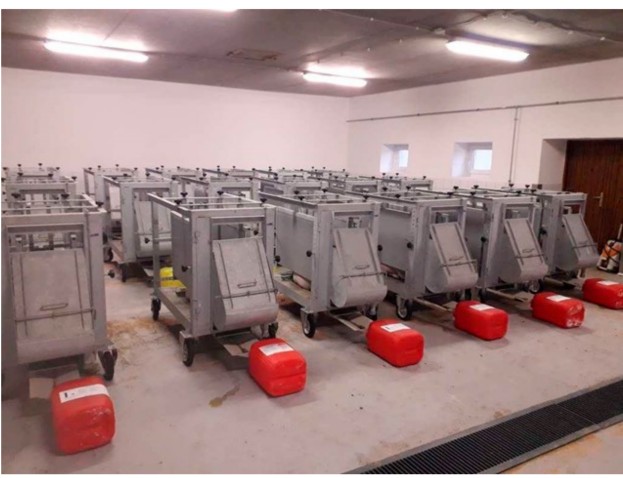

**Figure 1.** Balance cages used in the trial.

The balance experiment took 7 days. After 2 days and adaptation in the subsequent 5 days, the total amounts of feces and urine were collected daily and stored at $-10\ ^{\circ}C$. Before the analytical procedure, daily feces and urine samples from every pig were mixed and homogenized, and a representative sample of about 500 mL urine and 500 g feces was used for nitrogen analysis. To reduce the nitrogen loss from the urine, 20 mL of 5% sulfuric acid was poured into the urine containers. From the diets, feces and urine samples, their N contents were determined according to the Kjeldahl method with a Foss–Kjeltec 8400 Analyzer (Nils Foss Allé 1, DK-3400 Hilleroed, Denmark), and the most important N-balance parameters. were calculated. TAN percentage was calculated as the ratio of urinary N in the total N-excretion.

All data were analyzed using the SPSS 24.0 software. The general linear model and univariate ANOVA was used with genotype, live weight and dietary protein content as main factors. Levene's test was used to test the equality of error variances. If it was significant, the Brown–Forsythe test was used instead of F-probe. Linear regression was used to investigate the effect of N-intake on N-excretion within the genotypes and diet groups. Besides the significant effect of the main factors, their determination on the

investigated parameters was also investigated using the partial Eta squared values of the univariate ANOVA.

## 3. Results

No significant difference was found in the live weight of animals between the different treatment groups at the start of the N-balance experiment ($p > 0.05$) (Table 1).

**Table 1.** Initial average live weights of pigs at the start of the N-balance experiments (kg).

| Live Weight Categories | TD | | | HLW | | |
|---|---|---|---|---|---|---|
| | Control | T1.5 | T3 | Control | T1.5 | T3 |
| 20–30 kg | 26.70 | 26.55 | 26.75 | 27.63 | 27.27 | 26.67 |
| 30–40 kg | 37.35 | 37.42 | 36.77 | 35.30 | 34.28 | 34.05 |
| 40–80 kg | 61.92 | 61.43 | 61.57 | 63.60 | 60.70 | 59.90 |
| 80–110 kg | 91.70 | 91.83 | 91.75 | 93.24 | 89.48 | 88.84 |

TD: Topigs 20 × DanBred Duroc; HLW: Hungarian Large White; C: control diets; T1.5: Low protein diet with 1.5% protein reduction; T3: Low protein diet with 3% protein reduction.

The most important N-flow parameters are shown in Table 2. Focusing on the main factors, the nitrogen intake of pigs decreased in the low protein treatment groups. TD pigs ingested slightly more N than HLW pigs because TD pigs consumed more feed and their diets contained more protein. The higher feed intake of older pigs resulted in higher daily fecal N excretion. In spite of. significant differences found in the N-digestibility across the different live weight categories, these values were close to each other.

**Table 2.** N-flow parameters of fattening pigs as influenced by live weight, genotype and dietary protein content.

| Live Weight | Genotype | Diet | N Intake (g/Day) | Fecal N (g/Day) | N Digestibility (%) | Urinary N (g/Day) | TAN (%) | Total N Excretion (g/Day) | N Retention (%) |
|---|---|---|---|---|---|---|---|---|---|
| 20–30 kg | TD | C | 41.72 | 6.40 | 84.65 | 16.19 | 71.72 | 22.59 | 45.86 |
| | | T 1.5 | 39.12 | 5.68 | 85.49 | 13.68 | 70.10 | 19.36 | 50.62 |
| | | T 3 | 36.04 | 5.72 | 84.14 | 14.28 | 70.17 | 20.00 | 44.50 |
| | HLW | C | 41.25 | 5.88 | 85.74 | 8.88 | 59.44 | 14.76 | 64.23 |
| | | T 1.5 | 40.58 | 4.62 | 88.61 | 5.50 | 53.87 | 10.13 | 75.05 |
| | | T 3 | 30.58 | 4.86 | 84.11 | 3.77 | 43.68 | 8.63 | 71.79 |
| 30–40 kg | TD | C | 52.18 | 8.50 | 83.70 | 14.36 | 62.89 | 22.87 | 56.18 |
| | | T 1.5 | 46.04 | 6.71 | 85.44 | 11.52 | 62.96 | 18.23 | 60.30 |
| | | T 3 | 40.08 | 6.76 | 83.06 | 9.49 | 57.60 | 16.25 | 59.19 |
| | HLW | C | 48.66 | 8.76 | 81.99 | 11.22 | 55.95 | 19.99 | 58.93 |
| | | T 1.5 | 39.83 | 6.41 | 83.92 | 9.71 | 59.72 | 16.11 | 59.55 |
| | | T 3 | 37.09 | 6.33 | 82.94 | 5.60 | 46.25 | 11.93 | 67.83 |
| 40–80 kg | TD | C | 80.06 | 11.57 | 85.55 | 30.54 | 72.26 | 42.10 | 47.41 |
| | | T 1.5 | 72.44 | 13.80 | 80.95 | 19.47 | 57.98 | 33.27 | 54.07 |
| | | T 3 | 65.19 | 10.06 | 84.58 | 17.91 | 63.70 | 27.97 | 57.07 |
| | HLW | C | 62.88 | 10.27 | 83.67 | 14.32 | 58.18 | 24.59 | 60.89 |
| | | T 1.5 | 46.08 | 5.19 | 88.73 | 9.30 | 63.37 | 14.50 | 68.54 |
| | | T 3 | 51.36 | 8.27 | 83.91 | 9.30 | 52.39 | 17.57 | 65.79 |
| 80–110 kg | TD | C | 94.86 | 14.75 | 84.43 | 37.50 | 71.21 | 52.26 | 44.97 |
| | | T 1.5 | 85.48 | 13.13 | 84.64 | 32.71 | 70.34 | 45.84 | 46.37 |
| | | T 3 | 77.16 | 12.92 | 83.26 | 23.37 | 64.28 | 36.28 | 52.97 |
| | HLW | C | 81.78 | 19.01 | 76.75 | 23.97 | 55.10 | 42.98 | 47.44 |
| | | T 1.5 | 52.79 | 8.30 | 84.27 | 13.71 | 61.89 | 22.01 | 58.30 |
| | | T 3 | 47.60 | 7.55 | 84.15 | 15.67 | 67.41 | 23.21 | 51.24 |

**Table 2.** *Cont.*

| Live Weight | Genotype | Diet | N Intake (g/Day) | Fecal N (g/Day) | N Digestibility (%) | Urinary N (g/Day) | TAN (%) | Total N Excretion (g/Day) | N Retention (%) |
|---|---|---|---|---|---|---|---|---|---|
| Live weight | | 20–30 kg | 38.22 [d] | 5.53 [d] | 85.47 [a] | 10.38 [c] | 61.50 [ab] | 15.91 [c] | 58.67 [a] |
| | | 30–40 kg | 43.98 [c] | 7.25 [c] | 83.51 [b] | 10.32 [c] | 57.56 [b] | 17.56 [c] | 60.33 [a] |
| | | 40–80 kg | 63.00 [b] | 9.86 [b] | 84.56 [ab] | 16.81 [b] | 61.31 [ab] | 26.67 [b] | 58.96 [a] |
| | | 80–110 kg | 73.28 [a] | 12.61 [a] | 82.92 [b] | 24.49 [a] | 65.04 [a] | 37.10 [a] | 50.22 [b] |
| Diet | | C | 62.92 [a] | 10.64 [a] | 83.31 [b] | 19.62 [a] | 63.34 [a] | 30.27 [a] | 53.24 [b] |
| | | T 1.5 | 52.80 [b] | 7.98 [b] | 85.26 [a] | 14.45 [b] | 62.53 [a] | 22.43 [b] | 59.10 [a] |
| | | T 3 | 48.14 [c] | 7.81 [b] | 83.77 [b] | 12.42 [c] | 58.19 [b] | 20.23 [c] | 58.80 [a] |
| Genotype | | TD | 60.87 [a] | 9.67 [a] | 84.16 | 20.09 [a] | 66.27 [a] | 29.75 [a] | 51.63 [b] |
| | | HLW | 48.37 [b] | 7.95 [b] | 84.07 | 10.91 [b] | 56.44 [b] | 18.87 [b] | 62.47 [a] |
| | | Pooled SEM | 1.564 | 0.327 | 0.266 | 0.803 | 0.850 | 1.049 | 0.890 |
| *p*-Values | | | | | | | | | |
| Live weight (LW) | | | 0.0001 | 0.0001 | 0.0001 | 0.0001 | 0.0001 | 0.0001 | 0.0001 |
| Genotype (G) | | | 0.0001 | 0.0001 | 0.824 | 0.0001 | 0.0001 | 0.0001 | 0.0001 |
| Diet (D) | | | 0.0001 | 0.0001 | 0.001 | 0.0001 | 0.001 | 0.0001 | 0.0001 |
| GxLW | | | 0.0001 | 0.0001 | 0.001 | 0.0001 | 0.001 | 0.0001 | 0.0001 |
| GxD | | | 0.0001 | 0.0001 | 0.0001 | 0.276 | 0.042 | 0.016 | 0.479 |
| LWxD | | | 0.0001 | 0.0001 | 0.061 | 0.009 | 0.037 | 0.0001 | 0.413 |
| GxLWxD | | | 0.0001 | 0.0001 | 0.0001 | 0.013 | 0.001 | 0.001 | 0.05 |

[a–d] Mean values in one column within a main effect not sharing a common letter differ significantly ($p < 0.05$); TD: Topigs 20 × DanBred Duroc; HLW: Hungarian Large White; C: control diets; T1.5: Low protein diet with 1.5% protein reduction; T3: Low protein diet with 3% protein reduction.

As expected, groups consuming reduced protein diets had lower fecal N excretion values than the control group. However, no differences were found between the T1.5 and T3 treatments. The N-digestibility increased when low-protein diets were fed. Treatment T3 failed to cause further improvement compared with T1.5.

The higher fecal N-excretion of TD pigs was due to the higher feed and protein intake of these animals. However, no differences in the N-digestibility between the two genotypes were found.

Urinary N-excretion is responsible mostly for ammonia emissions. The daily excretion, similarly to the fecal N, increased with the age of pigs, but the TAN, which means the percentage of urinary N in the total N excretion, was similar in the first three fattening periods (57.6–61.5%) and increased afterwards, in the finisher phase (65%).

In both LP diet groups, the urinary N excretion decreased significantly. The average daily reduction was 5.17 and 7.2 g N/day in the T1.5 and T3 treatments, respectively. These numbers mean that the urinary N decreasing effect of LP diets is not linear. In this experiment, 1.5% and 3.0% dietary crude protein reduction resulted in 26.6% and 36.6% decrease in urinary N excretion respectively. Converting these numbers to a 1% dietary CP reduction means a 17.7% and 12.2% urinary N reduction in the T1.5 and T3 treatments, respectively (Figure 2).

The TAN was also affected by the CP content of diets. However, compared with the control, only treatment T3 resulted in significantly lower TAN. Significant genotype effect was observed in the urinary N. The excretion of TD pigs was higher, which resulted in also significantly higher TAN in these animals.

The daily total N excretion was affected by all three main factors. This parameter increased with age and CP content of diets. Due to the higher fecal and urinary excretion of TD pigs, the total N content was also in this genotype higher. The following figures demonstrate the regression between N-intake and N-excretion as affected by the genotype and the CP content of the diet (Figure 3). The slope of the regression equations is lower

in LP diets, and in the case of HLW pigs, this means one unit N-intake results in less N-excretion. The LP diet effects were more pronounced in the HLW genotype, when the 0.47 g N-excretion of control diets was reduced to 0.36 g and 0.37 g in the case of T1.5 and T3 treatments, respectively. Less LP diet effect was found in the case of TD animals (0.50, 0.49, and 0.46 g N excretion in the control, T1.5 and T3 treatments, respectively).

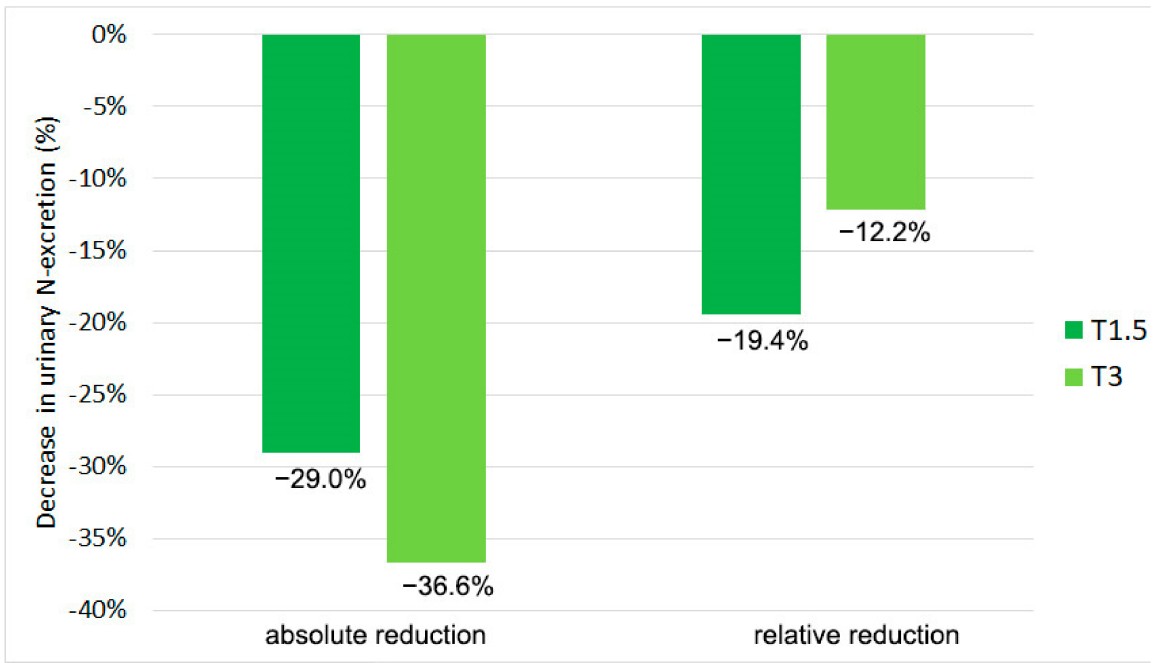

**Figure 2.** Effects of feeding LP diets on the reduction in urinary N excretion. T1.5: Low-protein diet with 1.5% protein reduction; T3: Low-protein diet with a 3% protein reduction.

In the grower period, no big difference in the N-retention of young pigs was found (58.7–60.3%). This parameter decreased significantly in the finisher phase (50.2%). Feeding LP diets improved the N-retention of the animals. In this trial, the improvement was about 5–6%. No difference between the two LP diets on N-retention was found.

In most cases, significant interactions have been found between the main factors. The reasons can be explained by the different trends between the genotypes, live weight or dietary protein responses (Table 2). Focusing only on the most important interactions, nitrogen digestibility was higher in the starter or grower II phases in HLW pigs, but the opposite was found in grower I or finisher periods. The response of the two genotypes on the change in dietary protein was also different. Dietary protein content failed to change the N-digestion of HLW pigs, but digestibility improved when LP diets were fed in TD pigs. TAN content of the total N excretion was usually higher in TD pigs; however, there were both dietary protein and live weight interactions. The differences between the two genotypes were bigger in the starter and finisher but smaller in the grower phases. Feeding LP diets reduced the TAN, but this was not the case for TD piglets in the starter phase. The N-retention was more favorable in HLW pigs. The differences were bigger in the starter and second grower phases but small in the remaining two phases. Lowering the protein content of the diets resulted in the highest N-retention increase in HLW pigs in the starter phase and TD animals in the second grower phase.

Beyond the significant effects, the determination of the main factors has also been evaluated. The test of between-subject effects revealed that the age of pigs had the greatest effect on the fecal N excretion ($r^2 = 0.812$) due to the increased protein uptake. After the age, the protein content of the diets was the second strong factor ($r^2 = 0.503$). In the case of N-digestibility, only weak effects of age ($r^2 = 0.162$) and dietary protein content ($r^2 = 0.122$) were found.

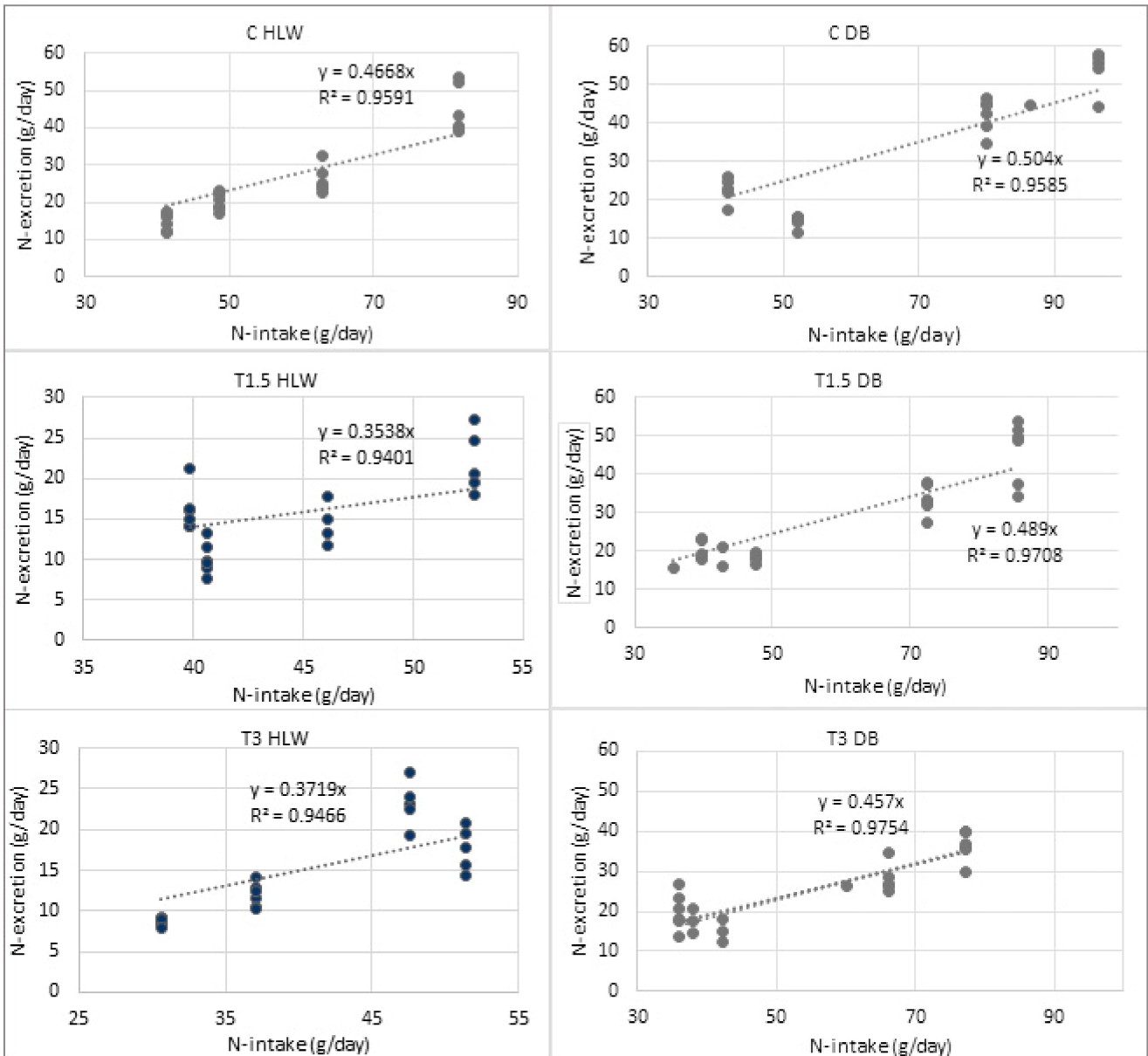

**Figure 3.** Regression between N intake and N excretion as influenced by genotype and the crude protein content of the diets. TD: Topigs 20 × DanBred Duroc; HLW: Hungarian Large White; C: control diets; T1.5: Low protein diet with 1.5% protein reduction; T3: Low-protein diet with 3% protein reduction.

The urinary N-excretion was mostly influenced also by age ($r^2 = 0.735$), but genotypes ($r^2 = 0.632$) and dietary treatments ($r^2 = 0.429$) played an important role in this parameter too. It follows from the above-mentioned determination that total N excretion was also mostly affected by age ($r^2 = 0.866$). For TAN and N retention, genotype had the greatest influencing effect, with $r^2$ values of 0.388 and 0.471, respectively. The N-retention of pigs was affected mostly by genotype ($r^2 = 0.471$) and age ($r^2 = 0.323$).

The total N excretion of pigs was also calculated on an annual level in order to compare with the default values of the EMEP/EEA [3]. The results are shown in Figure 4. The Tier 2 methodology is calculated currently with 12.1 kg N excretion/animal/year for fattening pigs, which is represented by the red line. This value corresponds only with the measured excretion of the older pigs of about 70 kg live weight and above. The N-excretion for the whole fattening of both genotypes is lower than the default value. Younger pigs excrete

significantly lower amounts of N and feeding LP diets also improves this trait. The protein content of the feed has a real potential to reduce the N-excretion. The annual reduction can be 2.8 and 3.6 kg N when 1.5 or 3% LP diets are fed.

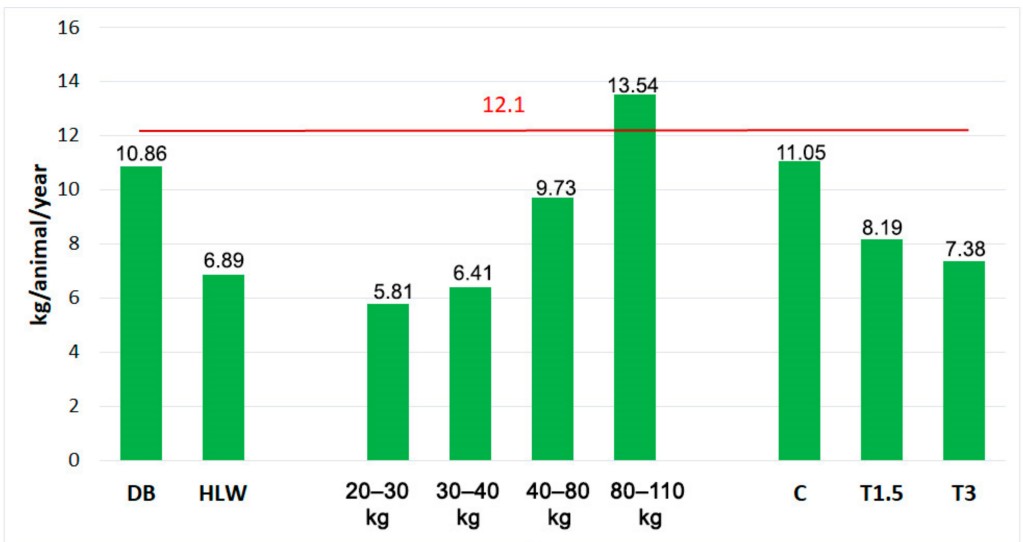

**Figure 4.** Total N-excretion of pigs as influenced by genotype, age, and dietary crude protein content (kg/animal/year). TD: Topigs 20 × DanBred Duroc; HLW: Hungarian Large White; C: control diets; T1.5: Low protein diet with 1.5% protein reduction; T3: Low protein diet with 3% protein reduction.

Among the nitrogen flow parameters, TAN had the highest standard deviation. At present, the TAN for pigs in all age groups and utilization types is estimated as 70% [3]. This value is close to that we have measured with the TD pigs (66%). However, in the case of Hungarian Large White animals this value is lower (56%) (Figure 5).

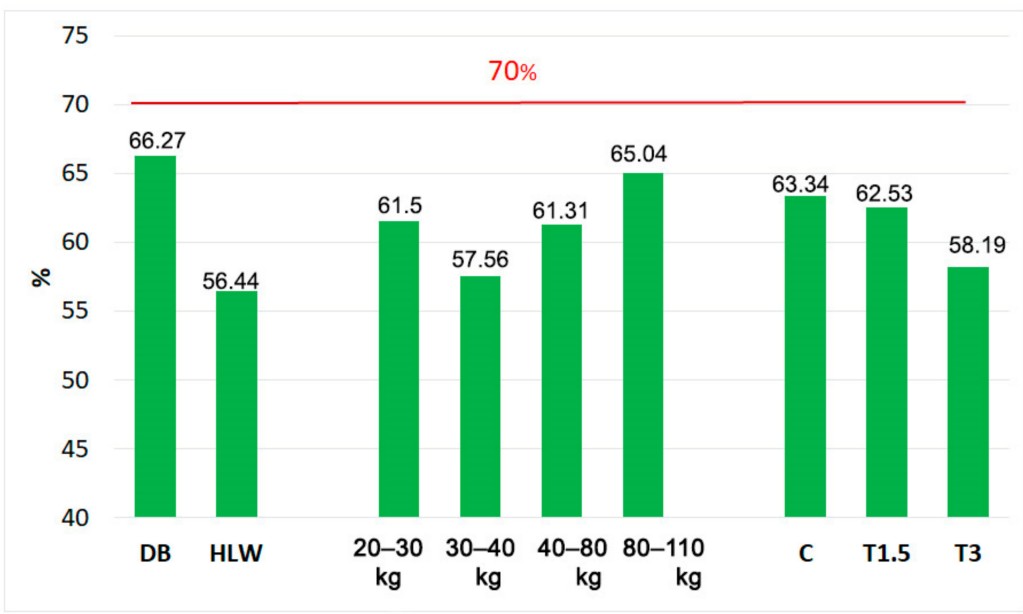

**Figure 5.** Total ammoniacal nitrogen (TAN) content of the total excreted nitrogen as influenced by genotype, age, and dietary crude protein content (%). TD: Topigs 20 × DanBred Duroc; HLW: Hungarian Large White; C: control diets; T1.5: Low protein diet with 1.5% protein reduction; T3: Low protein diet with 3% protein reduction.

Since both fecal and urinary N excretion increased with age, the TAN did not change significantly and was about 10% lower until 80 kg live weight than the default value. In the finisher phase, the TAN increased to 65%, but it is still less than the commonly used 70%. Decreasing the protein content of the diets had only a limited effect on TAN because CP reduction decreased both the fecal and urinary N excretion. Only the 3% protein reduction resulted in significant changes compared with the control.

## 4. Discussion

The results indicate that the N-excretion and TAN excretion values of the animals can be refined by considering age, genotype and feeding characteristics. Unfortunately, the effect of genotype on ammonia emission has been not widely studied so far. Better growth performance with higher protein deposition, in turn, means less nitrogen excretion and thus ammonia emissions in principle [2]. However, in our balance trial, HLW pigs of the semi-intensive genotype were more favorable from the N-and TAN excretion point of view, which means less ammonia emission. This can be explained at least partly with the higher dietary protein content and feed consumption of the crossbred genotype. Of course, it was not a fattening trial, and restricted feeding was used. Thus, the more intensive growth rate and reduced fattening period could compensate for the differences between the two genotypes.

Most of the previous studies used only one age category, which usually means animals weighing 50–60 kg. To our knowledge, there are no available scientific results on the comparison of the N-balance of different genotypes at different live weight categories.

Our urine, feces and total nitrogen excretion values are similar to those found in the literature for the 20–40 kg [7], 40–50 kg [10] or 60–110 kg live weight categories [11]. In agreement with others, only small differences exist between the N-digestion of different genotypes and age categories [11,12]. In our trial, only the N-digestibility of the starter diet was significantly higher. The reason could be that these starter diets contained hemoglobin, a highly digestible protein source of animal origin.

The improved N-digestibility of low-protein diets could be related to their higher crystalline amino acid content. The absorption of crystalline amino acids is close to 100%, significantly better than that of feed proteins [13,14]. There is no explanation why T3 treatment failed to cause further improvement in this parameter.

Generally, the balance experiments show that the decrease in N emissions is mainly due to a decrease in urinary nitrogen. Reducing the crude protein content of diets has a greater effect on urinary nitrogen than on fecal nitrogen excretion [15,16], as we have seen also in our experiment. Literature data suggest about 10% lower urinary N excretion and ammonia emissions when the dietary protein content is reduced by 1% [6,11,17]. One of our most interesting results is that the effect of protein reduction on ammonia emissions is not linear. The relative decrease is getting lower if the protein reduction is higher. This must be considered when the effect of the protein content of the diet is concerned. It can be important when the best available techniques (BAT) are defined.

Besides the genotypes, the age of animals had also a significant effect. The daily urinary N excretion of pigs below 40 kg was almost the same, about 10 g/day). The excretion significantly increased. afterward in each live weight category. So, using three live weight categories (<40 kg; 40–80 kg; 80 kg<) in the ammonia emission calculations seems to be accurate enough.

The total N-excretion and TAN content of the excreted N are more favorable than the default values of the different inventories. Therefore, using up-to-date research data in the national ammonia inventories could improve the quantity of ammonia emission.

Our retention results are more favorable compared to many previous publications [10,11] but are close to those of Figueroa et al. [7] and Gloaguen et al. [18]. The reason for the differences is the incorporation of more efficient new genotypes and the more crystalline amino acid incorporation into the diets.

Nitrogen retention measured in balance experiments is a more sensitive trait than weight gain or feed conversion. It is evident that younger pigs retain more N than older animals. Feeding LP diets with crystalline amino acid supplementation of lysine, threonine, tryptophan and methionine improve the utilization of dietary protein, which decreases urinary N loss. Similar to our findings, in most of the papers, improved N-retention can be found when LP diets are fed. Canh et al. ([11]) fed 16.5; 14.5 and 12.5% crude protein diets with male 55-kilogram live weight fattening pigs. Feeding the LP diets improved the N-retention of pigs from 39.1% (control diet) to 42.3 and 47.9% in the case of the 14.5 and 12.5% crude protein diets. Similarly, in the paper of Gloaguen et al. ([18]), the N-retention of the six-week-old Pietrain × Large White × Landrace pigs increased from 639 to 72.2% when fed an LP diet. However, there are also results when amino acid-supplemented LP diets significantly failed to affect N-retention [7]. In our case, the more intensive crossbred pigs retained less N in the balance cages. The potential reasons for this apparent inconsistency are unknown.

**5. Conclusions**

Based on our results, it can be stated that all the investigated main factors, the genotype, the age of pigs and the protein content of the diets have significant effects on the N-balance of fattening pigs. Among the N-balance traits, N-digestibility was hardly influenced. The urinary N ratio of the total excreted N, which is responsible mostly for the rate of ammonia emission from the manure, showed the highest variance in response to the treatments. The TAN content of excreted N until an 80-kilogram live weight is about 60% but increases afterward to 65%. Regarding the dietary treatments, compared with the control the TAN, was decreased only at a 3% protein reduction from 63.3% to 58.2%. Remarkably, a 10% difference was found in this trial between the TAN of the two different genotypes. The reason for this was mostly the significantly higher urinary N excretion of the TD pigs.

Feeding pigs an LP diet is one of the most efficient ways to mitigate ammonia emission. From our results, however, it can be concluded that the effect of dietary protein reduction is not linear. In the case of 1.5% dietary protein, a decrease of 19.4% TAN and ammonia emission reduction can be calculated for each percentage protein decrease. This decrease is lower (12.2%) at a 3% dietary protein reduction. It should be considered if the effects of LP diets on ammonia emissions are calculated.

The measured N-balance parameters—N-excretion, TAN-content of the excreted N and N-retention of pigs—are more favorable than the default values which can be found in the official recommendations. Using the more detailed, national, measured parameters can help reach the ammonia mitigation goals of the EU member states.

**Author Contributions:** Conceptualization, K.D. and I.A.G.-K.; methodology, I.A.G.-K., Z.B. and L.W.; formal analysis, I.A.G.-K. and N.S.; investigation, I.A.G.-K., Z.B., N.S., L.P., Á.K., Á.B. and J.P.; resources, K.D.; data curation, N.H.B., J.P. and I.A.G.-K.; writing—original draft preparation, I.A.G.-K.; writing—review and editing, K.D.; visualization, I.A.G.-K.; supervision, K.D.; project administration, N.H.B.; funding acquisition, K.D. and Z.B. All authors have read and agreed to the published version of the manuscript.

**Funding:** This research was supported by the Hungarian Ministry of Agriculture (AKGF/43/2021).

**Institutional Review Board Statement:** The animal experiment was approved by Food Chain Safety and Land Office Department of the Zala County Government Office under the license number ZAI/040/01010-7/2018.

**Informed Consent Statement:** Not applicable.

**Data Availability Statement:** All data generated or analysed during this study are included in this published article.

**Acknowledgments:** This work was supported by the Hungarian Government and the European Union by the EFOP-3.6.3-VEKOP-16-2017-00008 project. The project is cofinanced by the European Union and the European Social Fund.

**Conflicts of Interest:** All authors declare no conflict of interest.

## Appendix A

**Table A1.** Composition and nutrient content of experimental diets fed in the 20–30 kg live weight category (g/kg).

| Ingredients | TD | | | HLW | | |
|---|---|---|---|---|---|---|
| | C | T1.5 | T3 | C | T1.5 | T3 |
| Maize | 444 | 467 | 528 | 446 | 440 | 529 |
| Wheat | 240 | 260 | 238 | 279 | 290 | 235 |
| Soybean meal | 246 | 197 | 153 | 206 | 216 | 177 |
| Hemoglobin | 20 | 20 | 20 | 25 | 0.0 | 0.0 |
| Sunflower oil | 22 | 26 | 28 | 17 | 23 | 26 |
| L-Lysine | 0.8 | 2.1 | 3.3 | 0.8 | 3.0 | 4.2 |
| DL-Methionine | 1.0 | 1.2 | 1.4 | 1.1 | 1.2 | 1.4 |
| L-Threonine | 0.2 | 0.7 | 1.3 | 0.3 | 1.0 | 1.5 |
| L-tryptophan | 0.0 | 0.0 | 1.0 | 0.0 | 2.0 | 5.0 |
| Salt | 3.2 | 3.2 | 3.2 | 3.1 | 3.6 | 3.6 |
| Limestone | 12.2 | 12.4 | 12.5 | 11.2 | 11.0 | 11.2 |
| MCP | 5.8 | 6.0 | 6.4 | 5.8 | 6.1 | 6.5 |
| Premix | 5.0 | 5.0 | 5.0 | 5.0 | 5.0 | 5.0 |
| **Nutrient Content** | | | | | | |
| DE (MJ/kg) [1] | 14.19 | 14.20 | 14.01 | 14.14 | 14.25 | 14.15 |
| Dry matter [2] | 884.9 | 886.2 | 884.8 | 884.8 | 887.3 | 887.2 |
| Crude protein [2] | 194.6 | 185.7 | 168.1 | 175.4 | 159.1 | 147.0 |
| Crude fiber [2] | 29.9 | 29.6 | 28.2 | 26.3 | 27.9 | 25.4 |
| Crude fat [2] | 41.9 | 48.7 | 52.5 | 39.8 | 44.7 | 48.4 |
| Crude ash [2] | 46.4 | 47.0 | 43.4 | 44.3 | 43.9 | 42.4 |
| Nitrogen free extract [2] | 572.1 | 575.2 | 592.6 | 599.0 | 601.7 | 624.0 |
| Lysine [2] | 11.2 | 10.9 | 11.5 | 11.2 | 10.9 | 10.8 |
| Methionine [2] | 4.3 | 4.1 | 4.4 | 4.4 | 4.5 | 4.3 |
| Cystine [2] | 3.4 | 3.1 | 3.2 | 3.0 | 3.0 | 2.8 |
| Threonine [2] | 7.2 | 7.4 | 7.1 | 7.4 | 7.6 | 7.1 |
| Digestible Lysine [3] | 7.0 | 7.0 | 7.0 | 6.2 | 6.2 | 6.2 |
| Digestible Methionine [3] | 2.8 | 2.7 | 2.7 | 2.4 | 2.4 | 2.4 |
| Digestible Met + Cyst [3] | 4.8 | 4.6 | 4.5 | 4.4 | 4.2 | 4.0 |
| Digestible Threonine [3] | 4.5 | 4.6 | 4.5 | 4.1 | 4.1 | 4.1 |
| Digestible Tryptophan [3] | 1.4 | 1.2 | 1.2 | 1.3 | 1.3 | 1.3 |
| Ca [2] | 6.8 | 6.9 | 6.9 | 6.3 | 7.2 | 6.4 |
| P [2] | 5.0 | 4.9 | 4.4 | 5.0 | 4.9 | 4.8 |
| Available P [3] | 2.2 | 2.2 | 2.2 | 2.3 | 2.3 | 2.3 |
| Na [3] | 1.7 | 1.7 | 1.7 | 1.5 | 1.5 | 1.5 |

[1] calculated from the nutrients (DE (MJ/kg) = 0.0203 × crude protein + 0.0217 × crude fat + 0.0097 × crude fiber + 0.0158 × N-free extract; nutrients in g/kg); [2] measured parameters; [3] predicted form the measured nutrient contents of the ingredients. TD: Topigs 20 × DanBred Duroc; HLW: Hungarian Large White; C: control diets; T1.5: Low protein diet with 1.5% protein reduction; T3: Low protein diet with 3% protein reduction. Premix was supplied by UBM Ltd. (Pilisvörösvár, Pest County, Hungary); The active ingredients contained in the premix were as follows (per kg of diet): Ca 1.42 g; Na 0.35 g; vitamin A 6500 NE; vitamin D3 1500 NE; vitamin E 80 mg; Cu (E4) 21 mg; vitamin K3 2 mg; thiamine 1.5 mg; riboflavin 3 mg; pyridoxine 2 mg; cobalamin 15 μg; niacin 15 mg; pantothenic acid 10 mg; folic acid 0.25 mg; biotin 0.15 mg; choline chloride 125 mg; Mn (E5) 40 mg; Ca-iodide anhydrate 0.5 mg; Se 0.4 mg; Fe (sulfate) 81.5 mg; phytase EC 3.1.3.26 500 FTU; endo-xylanase 16,000 BXU.

**Table A2.** Composition and nutrient content of experimental diets, fed in the 30–40 kg live weight category (g/kg).

| Ingredients | TD | | | HLW | | |
|---|---|---|---|---|---|---|
| | C | T1.5 | T3 | C | T1.5 | T3 |
| Maize | 411 | 428 | 464 | 387 | 443 | 468 |
| Wheat | 331 | 360 | 366 | 351 | 332 | 350 |
| Soybean meal | 220 | 170 | 122 | 230 | 190 | 144 |
| Sunflower oil | 10 | 12 | 16 | 8.0 | 8.0 | 8.0 |
| L-Lysine | 3.1 | 44 | 57 | 0.9 | 021 | 34 |
| DL-Methionine | 1.1 | 1.3 | 1.5 | 0.6 | 0.8 | 1.0 |
| L-Threonine | 0.9 | 1.5 | 2.0 | 0.0 | 0.6 | 1.1 |
| L-tryptophan | 0.0 | 0.0 | 0.2 | 0.0 | 0.2 | 0.4 |
| Salt | 4.2 | 4.2 | 4.2 | 3.6 | 3.6 | 3.6 |
| Limestone | 11.6 | 11.9 | 12.1 | 10.8 | 10.8 | 11.1 |
| MCP | 1.7 | 1.8 | 2.1 | 3.7 | 4.2 | 4.5 |
| Premix | 5.0 | 5.0 | 5.0 | 5.0 | 5.0 | 5.0 |
| **Nutrient Content** | | | | | | |
| DE (MJ/kg) [1] | 14.02 | 14.01 | 13.95 | 13.92 | 13.89 | 13.83 |
| Dry matter [2] | 880.4 | 880.1 | 878.0 | 876.4 | 877.9 | 876.4 |
| Crude protein [2] | 179.2 | 163.5 | 144.8 | 174.8 | 151.8 | 138.0 |
| Crude fbier [2] | 28.2 | 29.6 | 28.3 | 28.9 | 27.2 | 26.7 |
| Crude fat [2] | 31.6 | 36.0 | 38.6 | 29.4 | 31.1 | 30.5 |
| Crude ash [2] | 44.7 | 41.7 | 39.9 | 45.3 | 43.1 | 41.7 |
| Nitrogen free extract [2] | 596.7 | 609.3 | 626.4 | 598.0 | 624.7 | 639.5 |
| Lysine [2] | 10.2 | 10.0 | 10.3 | 10.5 | 10.7 | 10.8 |
| Methionine [2] | 3.9 | 4.2 | 4.0 | 4.0 | 4.1 | 4.0 |
| Cystine [2] | 3.5 | 3.1 | 2.7 | 3.1 | 2.8 | 2.8 |
| Threonine [2] | 7.4 | 7.0 | 7.2 | 7.2 | 7.0 | 7.1 |
| Digestible Lysine [3] | 7.0 | 7.0 | 7.0 | 6.2 | 6.2 | 6.2 |
| Digestible Methionine [3] | 2.8 | 2.7 | 2.7 | 2.4 | 2.4 | 2.4 |
| Digestible Met + Cyst [3] | 4.8 | 4.6 | 4.5 | 4.4 | 4.2 | 4.0 |
| Digestible Threonine [3] | 4.5 | 4.6 | 4.5 | 4.1 | 4.1 | 4.1 |
| Digestible Tryptophan [3] | 1.4 | 1.2 | 1.2 | 1.3 | 1.3 | 1.3 |
| Ca [2] | 6.6 | 6.4 | 6.5 | 6.3 | 6.2 | 7.3 |
| P [2] | 3.7 | 3.6 | 4.4 | 4.5 | 4.6 | 4.4 |
| Available P [3] | 2.2 | 2.2 | 2.2 | 2.3 | 2.3 | 2.3 |
| Na [3] | 1.7 | 1.7 | 1.7 | 1.5 | 1.5 | 1.5 |

[1] calculated from the nutrients (DE (MJ/kg) = 0.0203 × crude protein + 0.0217 × crude fat + 0.0097 × crude fiber + 0.0158 × N-free extract; nutrients in g/kg); [2] measured parameters; [3] predicted form the measured nutrient contents of the ingredients. TD: Topigs 20 × DanBred Duroc; HLW: Hungarian Large White; C: control diets; T1.5: Low protein diet with 1.5% protein reduction; T3: Low protein diet with 3% protein reduction. Premix was supplied by UBM Ltd. (Pilisvörösvár, Pest County, Hungary); The active ingredients contained in the premix were as follows (per kg of diet): Ca 1.42 g; Na 0.35 g; vitamin A 6500 NE; vitamin D3 1500 NE; vitamin E 80 mg; Cu (E4) 21 mg; vitamin K3 2 mg; thiamine 1.5 mg; riboflavin 3 mg; pyridoxine 2 mg; cobalamin 15 µg; niacin 15 mg; pantothenic acid 10 mg; folic acid 0.25 mg; biotin 0.15 mg; choline chloride 125 mg; Mn (E5) 40 mg; Ca-iodide anhydrate 0.5 mg; Se 0.4 mg; Fe (sulfate) 81.5 mg; phytase EC 3.1.3.26 500 FTU; endo-xylanase 16,000 BXU.

**Table A3.** Composition and nutrient content of experimental diets, fed in the 40–80 kg live weight category (g/kg).

| Ingredients | TD | | | HLW | | |
|---|---|---|---|---|---|---|
| | C | T1.5 | T3 | C | T1.5 | T3 |
| Maize | 559 | 500 | 524 | 581 | 528 | 564 |
| Barley | 164 | 271 | 290 | 170 | 268 | 270 |
| Soybean meal | 247 | 191 | 143 | 220 | 168 | 125 |
| Sunflower oil | 5 | 10 | 12 | 5 | 10 | 12 |
| L-Lysine | 1.4 | 2.8 | 4.0 | 0.4 | 1.8 | 3.0 |
| DL-Methionine | 0.8 | 1.0 | 1.2 | 0.4 | 0.6 | 0.8 |
| L-Threonine | 0.4 | 1.0 | 1.5 | 0.0 | 0.6 | 1.1 |
| L-tryptophan | 0.0 | 0.0 | 0.2 | 0.0 | 0.2 | 0.4 |
| Salt | 4.1 | 4.1 | 4.1 | 3.6 | 3.6 | 3.6 |
| Limestone | 10.8 | 11 | 11.2 | 10.8 | 10.9 | 11.1 |
| MCP | 3.4 | 3.5 | 3.8 | 3.9 | 4.1 | 4.4 |
| Premix | 5.0 | 5.0 | 5.0 | 5.0 | 5.0 | 5.0 |
| **Nutrient Content** | | | | | | |
| DE (MJ/kg) [1] | 13.89 | 13.88 | 13.85 | 14.37 | 14.16 | 14.02 |
| Dry matter [2] | 875.4 | 876.2 | 878.9 | 910.1 | 902.1 | 889.6 |
| Crude protein [2] | 161.7 | 147.8 | 158.8 | 162.4 | 139.8 | 128.4 |
| Crude fiber [2] | 34.3 | 36.3 | 36.2 | 33.1 | 34.3 | 31.5 |
| Crude fat [2] | 29.7 | 34.6 | 35.0 | 33.1 | 34.9 | 38.3 |
| Crude ash [2] | 45.0 | 42.9 | 43.7 | 46.7 | 45.6 | 41.2 |
| Nitrogen free extract [2] | 587.7 | 600.7 | 616.2 | 634.8 | 647.5 | 650.2 |
| Lysine [2] | 9.5 | 9.7 | 9.3 | 10.1 | 9.7 | 9.9 |
| Methionine [2] | 3.7 | 3.8 | 3.6 | 3.7 | 3.8 | 3.8 |
| Cystine [2] | 3.3 | 2.9 | 2.9 | 3.0 | 2.4 | 2.5 |
| Threonine [2] | 6.6 | 6.4 | 6.4 | 6.8 | 6.7 | 6.5 |
| Digestible Lysine [3] | 7.0 | 7.0 | 7.0 | 6.2 | 6.2 | 6.2 |
| Digestible Methionine [3] | 2.8 | 2.7 | 2.7 | 2.4 | 2.4 | 2.4 |
| Digestible Met + Cyst [3] | 4.8 | 4.6 | 4.5 | 4.4 | 4.2 | 4.0 |
| Digestible Threonine [3] | 4.5 | 4.6 | 4.5 | 4.1 | 4.1 | 4.1 |
| Digestible Tryptophan [3] | 1.4 | 1.2 | 1.2 | 1.3 | 1.3 | 1.3 |
| Ca [2] | 6.3 | 6.5 | 6.6 | 6.4 | 6.8 | 6.6 |
| P [2] | 4.4 | 4.0 | 4.4 | 4.3 | 4.0 | 3.8 |
| Available P [3] | 2.2 | 2.2 | 2.2 | 2.3 | 2.3 | 2.3 |
| Na [3] | 1.7 | 1.7 | 1.7 | 1.5 | 1.5 | 1.5 |

[1] calculated from the nutrients (DE (MJ/kg) = $0.0203 \times$ crude protein + $0.0217 \times$ crude fat + $0.0097 \times$ crude fiber + $0.0158 \times$ N-free extract; nutrients in g/kg); [2] measured parameters; [3] predicted form the measured nutrient contents of the ingredients. TD: Topigs $20 \times$ DanBred Duroc; HLW: Hungarian Large White; C: control diets; T1.5: Low protein diet with 1.5% protein reduction; T3: Low protein diet with 3% protein reduction. Premix was supplied by UBM Ltd. (Pilisvörösvár, Pest County, Hungary); The active ingredients contained in the premix were as follows (per kg of diet): Ca 1.42 g; Na 0.35 g; vitamin A 6500 NE; vitamin D3 1500 NE; vitamin E 80 mg; Cu (E4) 21 mg; vitamin K3 2 mg; thiamine 1.5 mg; riboflavin 3 mg; pyridoxine 2 mg; cobalamin 15 μg; niacin 15 mg; pantothenic acid 10 mg; folic acid 0.25 mg; biotin 0.15 mg; choline chloride 125 mg; Mn (E5) 40 mg; Ca-iodide anhydrate 0.5 mg; Se 0.4 mg; Fe (sulfate) 81.5 mg; phytase EC 3.1.3.26 500 FTU; endo-xylanase 16,000 BXU.

**Table A4.** Composition and nutrient content of experimental diets, fed in the 80–110 kg live weight category (g/kg).

| Ingredients | TD | | | HLW | | |
|---|---|---|---|---|---|---|
| | C | T1.5 | T3 | C | T1.5 | T3 |
| Maize | 436 | 439 | 496 | 560 | 568 | 565 |
| Barley | 367 | 411 | 395 | 240 | 273 | 320 |
| Soybean meal | 169 | 118 | 72 | 173 | 127 | 79 |
| Sunflower oil | 5.0 | 7.0 | 8.0 | 5.0 | 7.0 | 9.0 |
| L-Lysine | 2.1 | 3.4 | 4.6 | 0.7 | 2.0 | 3.3 |
| DL-Methionine | 0.8 | 0.9 | 1.1 | 0.4 | 0.5 | 0.7 |
| L-Threonine | 0.6 | 1.2 | 1.7 | 0.0 | 0.6 | 1.2 |
| L-tryptophan | 0.0 | 0.1 | 0.3 | 0.0 | 0.2 | 0.4 |
| Salt | 4.2 | 4.2 | 4.1 | 3.6 | 3.6 | 3.6 |
| Limestone | 9.0 | 9.2 | 9.4 | 10 | 10 | 10.2 |
| MCP | 2.2 | 2.4 | 2.7 | 2.8 | 3.0 | 3.2 |
| Premix | 5.0 | 5.0 | 5.0 | 5.0 | 5.0 | 5.0 |
| **Nutrient Content** | | | | | | |
| DE (MJ/kg) [1] | 14.02 | 14.03 | 13.89 | 14.21 | 13.91 | 13.84 |
| Dry matter [2] | 885.6 | 884.2 | 879.8 | 900.6 | 880.6 | 879.8 |
| Crude protein [2] | 158.8 | 140.6 | 126.9 | 145.2 | 129.9 | 119.0 |
| Crude fiber [2] | 28.8 | 28.0 | 27.2 | 36.4 | 28.9 | 34.2 |
| Crude fat [2] | 27.2 | 29.5 | 30.7 | 30.3 | 33.1 | 37.1 |
| Crude ash [2] | 42.4 | 36.5 | 37.8 | 39.9 | 38.5 | 38.5 |
| Nitrogen free extract [2] | 628.4 | 649.6 | 657.2 | 648.8 | 650.2 | 651.0 |
| Lysine [2] | 8.2 | 8.3 | 8.1 | 8.6 | 8.6 | 8.7 |
| Methionine [2] | 3.2 | 3.3 | 3.4 | 3.3 | 3.4 | 3.5 |
| Cystine [2] | 2.9 | 2.6 | 2.5 | 2.7 | 2.5 | 2.3 |
| Threonine [2] | 5.9 | 5.8 | 6.0 | 6.0 | 6.1 | 5.8 |
| Digestible Lysine [3] | 7.0 | 7.0 | 7.0 | 6.2 | 6.2 | 6.2 |
| Digestible Methionine [3] | 2.8 | 2.7 | 2.7 | 2.4 | 2.4 | 2.4 |
| Digestible Met + Cyst [3] | 4.8 | 4.6 | 4.5 | 4.4 | 4.2 | 4.0 |
| Digestible Threonine [3] | 4.5 | 4.6 | 4.5 | 4.1 | 4.1 | 4.1 |
| Digestible Tryptophan [3] | 1.4 | 1.2 | 1.2 | 1.3 | 1.3 | 1.3 |
| Ca [2] | 5.6 | 5.5 | 5.6 | 5.9 | 5.6 | 5.6 |
| P [2] | 3.7 | 3.5 | 3.6 | 3.8 | 3.5 | 6.6 |
| Available P [3] | 2.2 | 2.2 | 2.2 | 2.3 | 2.3 | 2.3 |
| Na [3] | 1.7 | 1.7 | 1.7 | 1.5 | 1.5 | 1.5 |

[1] calculated from the nutrients (DE (MJ/kg) = 0.0203 × crude protein + 0.0217 × crude fat + 0.0097 × crude fiber + 0.0158 × N-free extract; nutrients in g/kg); [2] measured parameters; [3] predicted form the measured nutrient contents of the ingredients. TD: Topigs 20 × DanBred Duroc; HLW: Hungarian Large White; C: control diets; T1.5: Low protein diet with 1.5% protein reduction; T3: Low protein diet with 3% protein reduction. Premix was supplied by UBM Ltd. (Pilisvörösvár, Pest County, Hungary); The active ingredients contained in the premix were as follows (per kg of diet): Ca 1.42 g; Na 0.35 g; vitamin A 6500 NE; vitamin D3 1500 NE; vitamin E 80 mg; Cu (E4) 21 mg; vitamin K3 2 mg; thiamine 1.5 mg; riboflavin 3 mg; pyridoxine 2 mg; cobalamin 15 μg; niacin 15 mg; pantothenic acid 10 mg; folic acid 0.25 mg; biotin 0.15 mg; choline chloride 125 mg; Mn (E5) 40 mg; Ca-iodide anhydrate 0.5 mg; Se 0.4 mg; Fe (sulfate) 81.5 mg; phytase EC 3.1.3.26 500 FTU; endo-xylanase 16,000 BXU.

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
