# Peer review of "Impacts of Age, Genotype and Feeding Low-Protein Diets on the N-Balance Parameters of Fattening Pigs"

_agriculture, doi:10.3390/agriculture12010094_

Round 1

Reviewer 1 Report

Q1: The introduction part is too wordy, there is no focus on the content you want to study, the previous several paragraphs of content are completely unnecessary to write. I suggest the introduction need to be rewritten.

Q2: Please indicate whether the nutrients content in the table in the appendix is analytical or calculated.

Q3: In the material method, the description of the test environment is not very rigorous.

Could you provide the temperature, humidity and lighting procedures? Also, how about the ventilation of the room, I am very interested in the NH3 concentration in the room.

Q4: L141, What does the representative sample mean?

Q5: L119: Why 1.5% and 3%? Why is there no data regarding to growth performance? Whether the difference in test results is related to the difference in feed intake? I mean without adding a discussion of growth performance, it is only from the N-balance point of view, which is of little significance in production.

Q6:L139: Is it reasonable to store faeces at -10℃? I mean whether microbial metabolism will affect N-related indicators

Q7: L301 Why choose hemoglobin as feed material?

Q8 L302-303: Please provide related reference.

Q9:L315-L318: The discussion here is too simplistic to make a clear distinction between age and low protein diets.

Q10:L330-331: Please describe specifically the literature you mentioned.

Reviewer 2 Report

The main concern in this research relates to experimental design. Additional explanations or statistical analysis are needed. After additional clarifications, it is possible to correctly interpret/perceive the results, discussion and the conclusions.
Material and methods
When you say DanBred pigs, is it a pure LW line (or some other pure line) or is it a crossbreed? This should be clearly stated in the paper.
What was the gender ratio in the groups? Were the groups gender balanced? This is important and needs to be clarified.
Models for statistical analysis are not the clearest. It is necessary to state the equations of the model with an explanation of the factors involved.
Second, before using ANOVA, was the homogeneity of variance tested?
Line 149-150 Rephrase the sentence .... e.g. Linear regression effect of N-intake on N-excretion within the genotypes and diet groups ............. (similar, in that sense).
Results
Avoid empty sentences such as on lines 161-162.....and in the other places.

Round 2

Reviewer 2 Report

I thank the authors for the corrections. Below are two small remarks.

Since these are crossbreeds Topigs 20 and DanBred Duroc, they are not DanBred pigs. It could be referred to as the “crossbreed TD genotype” or some appropriate name.

What is the reason for using different mixtures for different genotypes? Please see tables A1-A4.
